# Preoperatively Treated Diffuse-Type Gastric Adenocarcinoma: Glucose vs. Other Energy Sources Substantially Influence Prognosis and Therapy Response

**DOI:** 10.3390/cancers13030420

**Published:** 2021-01-23

**Authors:** Ahmed A. Abdelhakeem, Xuemei Wang, Rebecca E. Waters, Madhavi Patnana, Jeannelyn S. Estrella, Mariela Blum Murphy, Allison M. Trail, Yang Lu, Catherine E. Devine, Naruhiko Ikoma, Prajnan Das, Brian D. Badgwell, Jane E. Rogers, Jaffer A. Ajani

**Affiliations:** 1Department of Gastrointestinal Medical Oncology, The University of Texas MD Anderson Cancer Center, Houston, TX 77030, USA; aaabdelhakeem@mdanderson.org (A.A.A.); mblum1@mdanderson.org (M.B.M.); atrail@mdanderson.org (A.M.T.); 2Department of Biostatistics, The University of Texas MD Anderson Cancer Center, Houston, TX 77030, USA; xuewang@mdanderson.org; 3Department of Anatomical Pathology, The University of Texas MD Anderson Cancer Center, Houston, TX 77030, USA; rwaters@mdanderson.org (R.E.W.); jsestrella@mdanderson.org (J.S.E.); 4Department of Abdominal Imaging, The University of Texas MD Anderson Cancer Center, Houston, TX 77030, USA; madhavi.patnana@mdanderson.org (M.P.); catherine.devine@mdanderson.org (C.E.D.); 5Department of Nuclear Medicine, The University of Texas MD Anderson Cancer Center, Houston, TX 77030, USA; ylu10@mdanderson.org; 6Department of Surgical Oncology, The University of Texas MD Anderson Cancer Center, Houston, TX 77030, USA; nikoma@mdanderson.org (N.I.); bbadgwell@mdanderson.org (B.D.B.); 7Department of Radiation Oncology, The University of Texas MD Anderson Cancer Center, Houston, TX 77030, USA; prajdas@mdanderson.org; 8Pharmacy Clinical Programs, The University of Texas MD Anderson Cancer Center, Houston, TX 77030, USA; jerogers@mdanderson.org

**Keywords:** FDG, PET, prognosis, diffuse gastric cancer

## Abstract

**Simple Summary:**

The diffuse type of gastric adenocarcinoma (dGAC) generally confers a poor prognosis compared to intestinal type. Some dGACs are not avid on fluorine-18 fluoro-2-deoxy-D-glucose PET (FDG-PET) while others seem to consume glucose avidly. We analyzed the outcomes based on the avidity of the primary on baseline FDG-PET. Our data suggest that if dGACs used glucose as an energy source then the prognosis was very poor while non-glucose sources improved prognosis. Multi-platform (including metabolomics) profiling of dGACs would yield useful biologic understanding.

**Abstract:**

Diffuse type of gastric adenocarcinoma (dGAC) generally confers a poor prognosis compared to intestinal type. Some dGACs are not avid on fluorine-18 fluoro-2-deoxy-D-glucose PET (FDG-PET) while others seem to consume glucose avidly. We analyzed the outcomes based on the avidity (high with standardized uptake value (SUV) > 3.5 or low with SUV ≤ 3.5) of the primary on baseline FDG-PET. We retrospectively selected 111 localized dGAC patients who had baseline FDG-PET (all were treated with preoperative chemotherapy and chemoradiation). FDG-PET avidity was compared with overall survival (OS) and response to therapy. The mean age was 59.4 years and with many females (47.7%). The high-SUV group (58 (52.3%) patients) and the low-SUV group (53 (47.7%) patients) were equally divided. While the median OS for all patients was 49.5 months (95% CI: 38.5–98.8 months), it was 98.0 months (95% CI: 49.5–NE months) for the low-SUV group and 36.0 months for the high-SUV (*p* = 0.003). While the median DFS for all patients was 38.2 months (95% CI: 27.7–97.6 months), it was 98.0 (95% CI: 36.9–NE months) months for the low-SUV group was and only 27.0 months (95% CI: 15.2–63.2 months) for the high-SUV group (*p* = 0.005). Clinical responses before surgery were more common in the low-SUV group but overall we observed only 4 pathologic complete responses in 111 patients. Our unique data suggest that if dGACs used glucose as an energy source then the prognosis was very poor while non-glucose sources improved prognosis. Multi-platform (including metabolomics) profiling of dGACs would yield useful biologic understanding.

## 1. Introduction

Gastric adenocarcinoma (GAC) is the third leading cause of cancer-related deaths and ranks as the fifth most common cancer worldwide [1]. GAC represents 95% of all types of gastric cancers and is divided into two major subtypes: intestinal type (iGAC) and diffuse type (dGAC) with or without signet ring cells (SRCs) [2,3]. dGAC was relatively less frequently seen (32%) in the past compared to iGAC (54%) but its incidence has been rising alarmingly [4]. It is more frequently seen in females, younger individuals, and Blacks [4,5]. dGAC lacks the intercellular adhesion molecules and therefore, GAC cells do not form glands and often are dispersed as single cells or small clusters surrounded by fibrous stroma [4]. SRCs are unique in that they have abundant intra-cytoplasic mucin resulting in a nucleus shift near the cell wall giving it a “ring” appearance [6]. Little is known about SRCs in terms of their molecular biology; however, the presence of SRCs is associated with poor prognosis [7,8,9,10,11,12,13,14,15,16]. Only a small proportion (1–3%) of dGACs has been linked to germline mutations of *CDH1* leading to hereditary cancers [16]. For initial GAC staging, Fluorine-18 fluoro-2-deoxy-D-glucose PET-computed tomography (FDG-PET/CT) has no established role in GAC but it may be helpful to detect occult metastases [17]. FDG-PET/CT might provide additional information about the primary GAC that may be of use to clinicians and operating surgeons as dGACs tend to extend submucosally [18]. FDG-PET/CT may be also useful in differentiating dGAC subtypes [19] as the prognosis of dGAC patients is varied. The potentially low FDG avidity in dGAC is attributed to several factors, including the low-density diffuse infiltration of GAC cells, existence of extracellular or intracellular metabolically inert mucus content, use of glutamine as fuel rather than glucose, and low expression level of glucose transporter 1 (GLUT-1) [20,21]. Low avidity on PET has been reported as a beneficial prognosticator in dGAC patients who had primary surgery [22]. In the West, preoperative therapy is commonly given for localized GACs (including dGACs), we chose to determine the value of baseline FDG-PET in localized dGACs treated preoperatively as reported before [23]. In this study, we assessed the correlation between FDG-PET/CT avidity at baseline (prior to any treatment) and survival of preoperatively treated dGAC patients. 

## 2. Patients and Methods

We retrospectively identified localized dGAC patients treated at the University of Texas MD Anderson Cancer Center from our prospectively maintained databases between January 2005 and February 2019. Selected patients had an untreated localized (stage I to III) dGAC and a baseline PET-CT along with other staging to include endoscopic ultrasonography, laparoscopy, and blood work. Additionally, we collected demographics (age, sex, and ethnicity), Eastern Cooperative Oncology Group (ECOG) performance status, baseline body mass index (BMI), histologic grade and histologic subtype by the Lauren’s classification. Nearly all patients had chemotherapy and chemoradiation prior to surgery as previously reported [23]. Treatment and outcomes included overall survival (OS), progression-free survival (PFS) from the start of treatment, and best response. In each patient, we reviewed the standardized uptake values (SUV) maximum of the primary tumor. No other selection criteria were used.

### Statistical Analysis

OS was defined as the time interval between the first treatment date and death date or the surviving patients were censored at the last follow-up date. Disease-free survival (DFS) was defined as the time from the first treatment date to first progression date or death date, whichever came first, or the patients without progression were censored at the last follow-up date. Clinical Complete Response (cCR) was defined as resolution of the primary lesion including nodes on PET-CT or no evidence of progression on CTs along with negative endoscopic biopsy as described previously [8] but localized GACs are not amenable to standard response criteria used for advanced GACs [24,25].

The Kaplan–Meier method was used to estimate OS/DFS. The Fisher’s exact test was used to assess the difference in pathologic response rate between the high vs. low SUV groups. Statistical analyses were performed using SAS 9.4.(The SAS Institute, Cary, NC, USA).

The pathologic complete response (pCR) was defined as having no residual cancer cells in the entire specimen (primary and nodes) [11].

## 3. Results

A total of 111 patients were included in this project. Demographics and clinical characteristics are summarized in Table 1. The mean age was 59.4 years, with a large female cohort (47.7%). Most patients had an ECOG performance status of 1 (56.8%) or 0 (38.7%) while the remaining 4.5% had ECOG status equal to >1. All patients had dGAC. Histopathology confirmed that 100 (90%) patients’ dGAC contained SRCs, 10 (9%) were not specified, and only one case (0.9%) was mixed mucinous type with SRCs.

Patients were classified into two groups according to the level of SUV avidity of the primary lesion in the baseline PET-CT: the high-SUV (>3.5) group included 58 (52.3%) patients (Figure 1)**,** and the low-SUV (≤3.5) group included 53 (47.7%) patients (Figure 2).

Forty-three (81.1%) patients in the low-SUV group had a cCR, six (11.3%) patients had some response while the remaining four (7.6%) patients had either stable or progressive dGAC. Among the high-SUV group of patients, 33 (56.9%) patients had cCR, 14 (24.1%) patients had some response but 11 patients (18.8%) had either stable or progressive dGAC. Despite having a higher rate of cCR among the low-SUV group compared to the high-SUV group (81.1% vs. 56.9%, *p* = 0.008), the difference in the rate of any response was not significant (92.5% vs. 81.0%; *p* = 0.10).

The median OS for the entire cohort was 49.5 months (95% CI: 38.5–98.8 months) Figure 3.

The median OS for the low-SUV group was 98.0 months (95% CI: 49.5, not estimable (NE) months) and that for the high-SUV group was only 36.0 months (*p* = 0.003) (Figure 4).

The median DFS for the entire cohort was 38.2 months (95% CI: 27.7–97.6 months) The median DFS for the low-SUV group was 98.0 months (95% CI: 36.9–NE months) compared to only 27.0 months (95% CI: 15.2–63.2 months) for the high-SUV group (*p* value = 0.005). Of the 100 patients with SRCs in untreated samples, 48 patients were in the low-SUV group and 52 patients were in the high-SUV group. Post-treatment pathological specimens were assessed for SRC enrichment. Thirty (62.5%) patients in the low-SUV group and 29 (55.76%) patients in the high-SUV group had persistent SRCs. Further analysis of post-treatment SRC in the low-SUV group based on AJCC 8th edition [26] showed that a higher SRC percentage was noted when the stage (ypTNM) was higher than ypT1N0M0 and when there was regional lymph node(s) metastasis compared to the ypT0N0M0 and ypT1aN0M0 groups (Figure 5). The findings were the same for the high-SUV group **(**Figure 6). Only four (3.6%) patients achieved a pCR. One (1.9) pCR patient was in the low SUV group and three (5.3%) were in the high SUV group (not significant). 

## 4. Discussion

GAC is a major health burden with >1 million new cases/year globally. In the USA, 26,240 new cases are expected in 2020 [14,27,28]. It is the leading cause of cancer deaths in 10 countries [29]. Most patients are diagnosed in the late stages [14] and have an OS of <12 months. dGACs are poorly differentiated with the worst OS [3,14]. Others and our group have previously reported that dGACs are associated with resistance to therapy and short OS [10,11,12,13,30,31]. The incidence of poorly differentiated tumors and dGACs has been rising [4]. It is interesting that dGAC incidence has been rapidly rising in Blacks for reasons that are entirely unclear [4]. Therapeutic options are limited and mainly empiric [14]. Nevertheless, GACs are heterogenous and even dGACs can have varied outcomes. Clinical variables are not discriminating of the outcome but at the same time, we must acknowledge that we do not have different therapies for those who are likely to fare better and those who are likely to have rather rapid progression.

PET-CT is a useful tool and has shown promise in esophageal cancer [32]. The utility of PET-CT in GAC as a whole is not established. Some dGACs are known not to be avid on PET-CT at the outset. A decreased number of GLUT1 receptors may be one of the reasons but also some adenocarcinoma cells reprogram their metabolism to utilize glutamine as an energy source. What disadvantage glutamine might provide is under study. However, it is not clear if such a switch in the metabolism can change the clinical responsiveness and prognosis of the patients. There are no easy answers to such questions. We have asked a very simple question in which we tried to correlate the baseline SUV on PET-CT with prognosis and response to therapy. Our results may not be considered unique but they are certainly intriguing. There are no reports in the literature that are focused solely on these questions for preoperatively treated dGACs. We found that the prognosis of patients in the low SUV group was rather excellent (median OS = 98 months) but patients in the high SUV group did poorly (meaning glucose as an energy source improved fitness of cancer cell over the use of glutamine). Perhaps the next step could be to learn the biology of these groups at the molecular level.

The major drawback of our study is the retrospective nature of the project and another limiting but essential factor is that we could select only those dGAC patients who had baseline PET-CT. We were not able to balance patients in the low and high SUV groups. A validation of our finding is desired. There are only two studies that seem to address the role of PET-CT in GAC. An excellent study by Chon H et al. [22]. focused on 727 localized GAC patients who underwent surgery as primary therapy but all had baseline FDG-PET [22]. They also found that in the subset of patients with dGAC or SRC, high SUV leads to poor OS but the SUV avidity does not affect outcome in well/moderately differentiated GACs, even poorly differentiated GACs, or iGACs. Their results in dGACs/SRCs mimic ours except in the context that all our patients were pre-treated and most received chemotherapy followed by chemoradiation before surgery. Their study and our results convey a combined message that the altered metabolism of dGAC plays a role throughout the course of illness and is a determinant of patient prognosis irrespective of the treatment strategy offered. Meaning, the aggressive phenotype of glucose consuming dGAC remains unaltered whether they have extensive preoperative therapy (our population) or they undergo surgery first and then receive adjuvant therapy [22]. Arslan E et al. [33] studied 339 patients who had FDG-PET at baseline; however, only 102 patients were surgical candidates. Patients with SRC had low SUV plus this group had larger tumors, more frequent lymph node metastases. They did not provide prognostic data. 

## 5. Conclusions

In conclusion, we demonstrated that in 111 localized dGAC patients who were mostly treated with chemotherapy followed by chemoradiation then surgery, high FDG SUV (glucose as a source of energy) conferred poor prognosis. All dGACs appear to have inherent therapy resistance and aggressive clinical behavior. Molecular biology of patients with dGAC with high or low SUV might shed light on how to strategize therapy in the future.

## Figures and Tables

**Figure 1 cancers-13-00420-f001:**
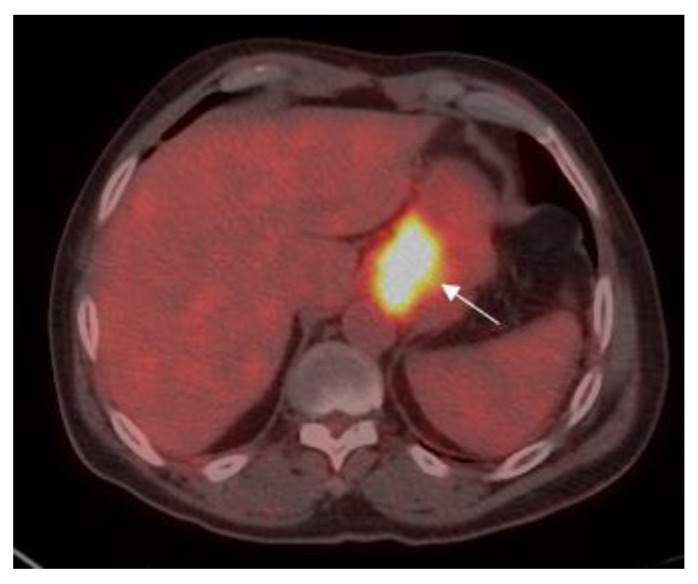
A 70-year old male with gastric adenocarcinoma (SRCC). Axial PET-CT images show low ^18^F-fluorine-18 fluoro-2-deoxy-D-glucose (FDG) uptake (standardized uptake value (SUV)_max_: 16.2) in primary tumor in the body of the stomach (white arrow).

**Figure 2 cancers-13-00420-f002:**
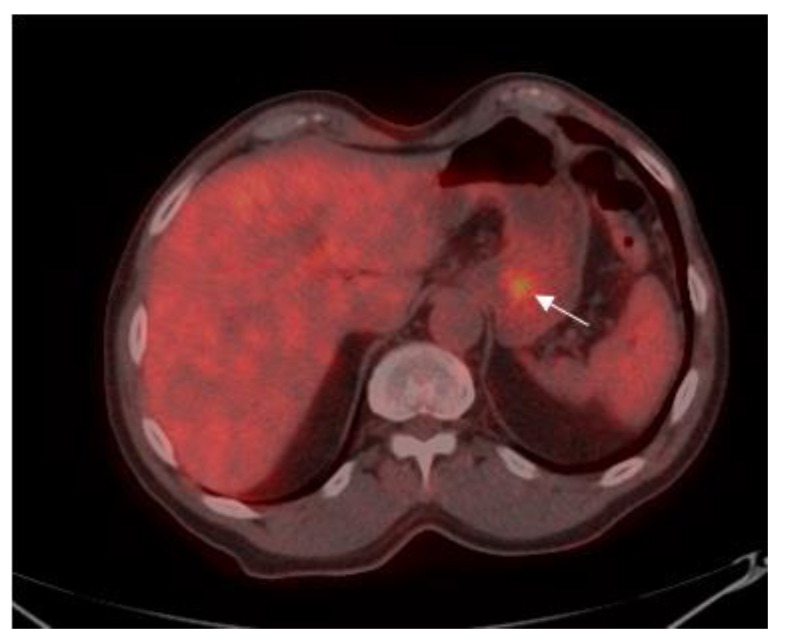
75-year old male with gastric adenocarcinoma (SRCC). Axial PET-CT images show low ^18^F-FDG uptake (SUV_max_: 3.5) in primary tumor in the body of the stomach (white arrow).

**Figure 3 cancers-13-00420-f003:**
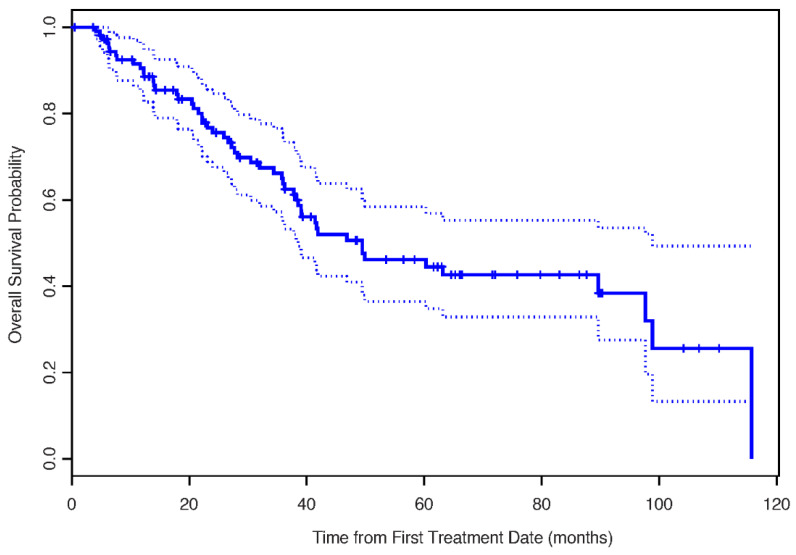
The Kaplan–Meier estimates for overall survival.

**Figure 4 cancers-13-00420-f004:**
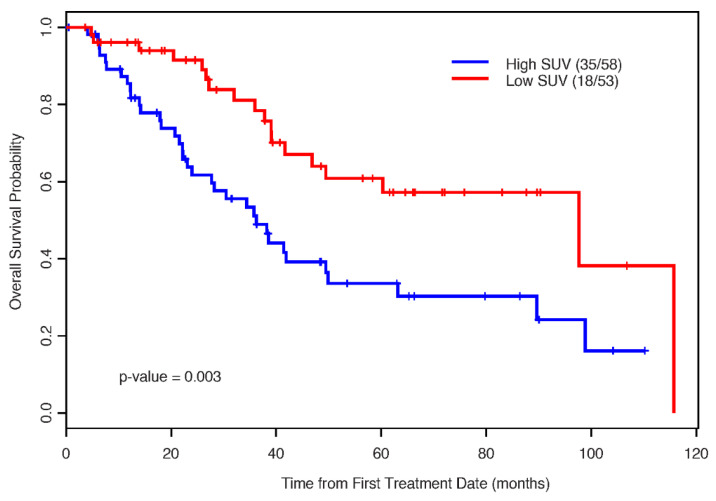
The Kaplan–Meier estimates for overall survival by SUV status.

**Figure 5 cancers-13-00420-f005:**
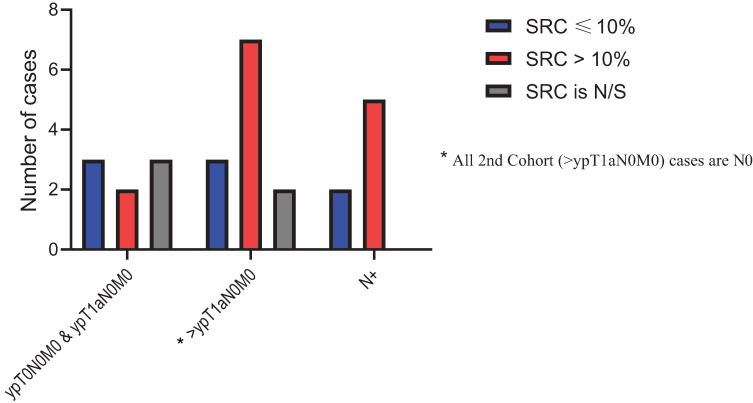
Post treatment signet ring cells (SRC) percentage distribution in Low-SUV group.

**Figure 6 cancers-13-00420-f006:**
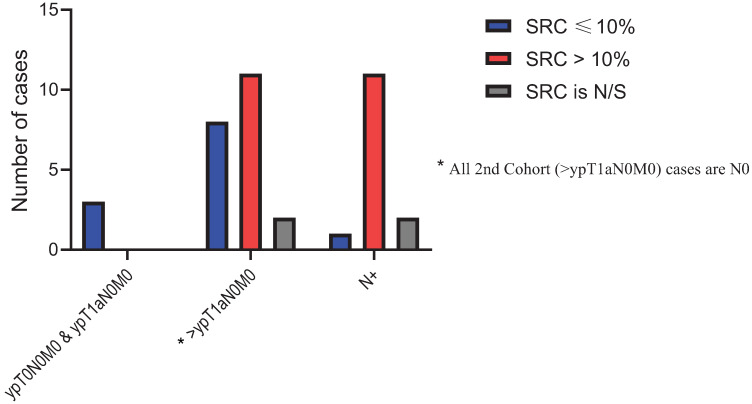
Post treatment SRC percentage distribution in High-SUV group.

**Table 1 cancers-13-00420-t001:** Demographic and clinical characteristics of patients.

Clinical Feature	Low SUV Group (*N*, %)	High SUV Group (*N*, %)
Total Number of cases	53	58
Age (mean ± SD)	59.5 ± 11.56	59 ± 13.94
Race		
White or Caucasian	29 (54.7%)	28 (48.3%)
Black or African American	6 (11.3%)	7 (12%)
Asian	6 (11.3%)	4 (6.9%)
Other	12 (22.7%)	19 (32.8%)
Sex		
Male	23 (43.4%)	35 (60.3%)
Female	30 (56.6%)	23 (39.7%)
Performance Status (ECOG)		
0	24 (45.3%)	19 (32.8%)
1	28 (52.8%)	35 (60.3%)
2	1 (1.9%)	3 (5.2%)
3	0	1 (1.7%)
Location of tumor		
Antrum	13 (25%)	14 (24.6%)
Antrum and Pyloric Canal	0 (0.0%)	1 (1.8%)
Body	20 (38.5%)	15 (26.3%)
Body and Antrum	2 (3.8%)	1 (1.8%)
Cardia	14 (26.9%)	20 (35.1%)
Cardia, Body and Antrum	2 (3.8%)	3 (5.3%)
Fundus	1 (1.9%)	0 (0.0%)
Fundus and Body	0 (0.0%)	1 (1.8%)
Pyloric Canal	0 (0.0%)	2 (3.5%)
Tumor differentiation		
Poorly differentiated	53 (100%)	58 (100%)
Adenocarcinoma subtype		
Signet Ring Carcinoma	48 (90.5%)	53 (91.4%)
Not Signet Ring Carcinoma	5 (9.5%)	5 (8.6%)
Tumor Histology		
Diffuse	53 (100%)	58 (100%)
Baseline T stage		
Tis	0	1 (1.7%)
T1	12 (22.6%)	1 (1.7%)
T2	15 (28.3%)	9 (15.5%)
T3	25 (47.2%)	39 (67.3%)
T4	1 (1.9%)	8 (13.8%)
Baseline N stage		
N0	40 (75.5%)	31 (53.5%)
N1	9 (16.9%)	16 (27.6%)
N2	2 (3.8%)	8 (13.7%)
N3	2 (3.8%)	3 (5.2%)
Baseline M		
M0	53 (100%)	53 (1005)
Baseline clinical stage		
0	0	1 (1.7%)
I	25 (47.2%)	8 (13.8%)
IIA	2 (3.8%)	2 (3.5%)
IIB	15 (28.3%)	22 (37.9%)
III	11 (20.7%)	25 (43.1%)
Response to First-Line Therapy		
Complete response	43 (81.1%)	33 (56.9%)
Partial response	6 (11.3%)	14 (24.1%)
Stable disease	1 (1.9%)	2 (3.5%)
Progressive disease	3 (15.5%)	9 (15.5%)

NOS, not otherwise specified; ECOG, Eastern Cooperative Oncology Group.

## Data Availability

The data presented in this study are available on request from the corresponding author.

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
