# Peer review of "Preoperatively Treated Diffuse-Type Gastric Adenocarcinoma: Glucose vs. Other Energy Sources Substantially Influence Prognosis and Therapy Response"

_cancers, 2021, doi:10.3390/cancers13030420_

Round 1
Reviewer 1 Report
In the present manuscript “Preoperatively Treated Diffuse-Type Gastric Adenocarcinoma: Glucose vs. Other Energy Sources Substantially Influence Prognosis and Therapy Response”, Abdelhakeem et al analyzed the prognostic role of FGD-update in diffuse gastric cancer. Patients with preoperative FDG avid gastric adenocarcinoma demonstrated significant worse outcome when compared to PET negative tumors in this retrospective study analyzing 111 patients. This is an interesting manuscript with regard to the fact, that a special and rare type of gastric cancer has been analyzed.
However, some questions came up after reading this otherwise well prepared manuscript.
Comment 1: How was baseline FDG defined, was it pre-chemo or pre-op (so was it in treatment naïve patients)? This is not completely clear to me at least after the abstract.
Comment 2: Why was 3.5 chosen as cut off for dividing into the PET positive and negative group? Should be discussed or explained in the methods section
Comment 3: How was response defined? It is stated in the methods section that the standard criteria are not applicable for localized GACs so one might wonder how the response was defined for this subgroup (which is also not represented in the present manuscript). In general, this sentence is a little bit of confusing to me. Please clearly focus on dGAC and define the response criteria. lGAC is not relevant in the methods section
Comment 4: Was response always metabolic or radiologic response? Was there always an initial PET CT, than chemotherapy, than re-evaluation by PET CT and then surgery? How many patients had to be excluded due to poor response to induction therapy? (also see next comment considering consort diagram)
Comment 5: A consort diagram summarizing study in and exclusion would help the reader getting a quick overview about the analyzed study population
Comment 6: OS was distinct compared to dfs in this study cohort, indicating, that even after recurrence, survival can be prolonged with modern oncology (50 vs 38 months). Thus, the question arises which adjuvant and other postoperative treatments were applied? Was SUV max correlating e.g. with the presence of promising treatment targets including PD-L1 and her2? Do you have any data on this issue?
Comment 7: in general, the metabolic response (after induction therapy) might be of interest besides overall FDG uptake/SUV max. Have you analyzed this data? Also compare and discuss the MUNICON trial
Author Response
December 26, 2020
Dear Editor:
We wish you thank you and your reviewers for the time and effort. We are grateful for the opportunity to respond to the queries raised by each reviewer. We are able to address each query. We believe the revised manuscript is now much better and clear. The point-by-point responses are listed below:
Comment 1: How was baseline FDG defined? Was it pre-chemo or pre-op (so was it in treatment naïve patients)? This is not completely clear to me, at least after the abstract.
Answer: Baseline FDG-PET/CT was defined as the staging PET/CT images obtained in treatment naïve localized diffuse-type gastric adenocarcinoma (dGAC) patients.
Comment 2: Why was 3.5 chosen as cut off for dividing into the PET positive and negative group? Should be discussed or explained in the methods section.
Answer: With consultation with the two radiology authors (YL and MP), a cutpoint of SUV of 3.5 was chosen. This was strictly based on the physiologic SUV uptake range used by the radiologists (3.5 is considered an acceptable cut-point; however, occasionally, <4.0 is considered acceptable). We believe 3.5 is generally an acceptable cut-point to discriminate normal SUV from abnormal SUV.
Comment 3: How was the response defined? It is stated in the methods section that the standard criteria are not applicable for localized GACs so one might wonder how the response was defined for this subgroup (which is also not represented in the present manuscript). In general, this sentence is a little bit of confusing to me. Please clearly focus on dGAC and define the response criteria. lGAC is not relevant in the methods section
Answer: Although we have published the definition of cCR several times, it was most recently described in by our group in Ann Surgery August 2020, PubMed ID 32675544. A clinical complete response (cCR) is determined approximately 6 weeks after the completion of chemoradiation, and at that time, the endoscopic biopsies should not have any evidence of cancer, and the PET‐CT SUV should be in the physiologic range or no evidence of progressive cancer on CTs (if PET-CT was not done).
Comment 4: Was response always metabolic or radiologic response? Was there always an initial PET CT, then chemotherapy, than re-evaluation by PET-CT and then surgery? How many patients had to be excluded due to poor response to induction therapy? (also see next comment considering consort diagram)
Answer: In this study, we were entirely focused on the value of baseline SUVmax of the primary diffuse type of GAC. All 111 patients underwent chemotherapy then chemoradiation (with a preoperative staging approximately 6 weeks after the completion of chemoradiation). We then assessed the rate of cCR, pathologic complete response rate (in the surgical specimens), and overall and disease-free survival. Therefore, the response assessment was not metabolic (which would have required a second PET-CT after chemoradiation in all patients, but we did not have this, and the focus was on the value of baseline (staging) PET-CT.
The number of patients excluded from surgery is due to poor response to induction therapy was only 9 (2 among the Low SUV group and 7 among the High SUV group).
Comment 5: A consort diagram summarizing study in and exclusion would help the reader getting a quick overview about the analyzed study population
Answer: We are providing a flow chart of 1,086 patients (as our denominator) that explains how we ended up with 111 patients. We hope this is helpful.
Comment 6: OS was distinct compared to dfs in this study cohort, indicating, that even after recurrence, survival can be prolonged with modern oncology (50 vs 38 months). Thus, the question arises which adjuvant and other postoperative treatments were applied? Was SUV max correlating e.g. with the presence of promising treatment targets including PD-L1 and her2? Do you have any data on this issue?
Answer: This is truly an interesting question, but we are not able to fully answer. The way we treat localized GAC (whether dGAC or intestinal GAC) is the way described in the manuscript and previously in JCO (PubMed ID 26324361). Patients receive chemotherapy (2 months), chemoradiation (5 weeks with 45 Gy), and then surgery. If the surgery is successful, these patients do not receive any further therapy irrespective of the assessed risk from the ypStabe. Therefore, Her2 or PD-L-1 status was not assessed or acted upon.
Comment 7: in general, the metabolic response (after induction therapy) might be of interest besides overall FDG uptake/SUV max. Have you analyzed this data? Also compare and discuss the MUNICON trial.
Answer: The reviewer is entirely accurate about the metabolic response; however, (please also see our answer to query no. 4). Our research plan or hypothesis did not include the assessment of metabolic response. We, nevertheless, appreciate the question and will review answering it in the future.
Please see our study data in the attachment section.
Finally, we are very appreciated. We hope these are satisfactory answers, and the manuscript is now acceptable for publication.
Sincerely,
Jaffer A Ajani

Reviewer 2 Report
The authors of this manuscript “Preoperatively Treated Diffuse-Type Gastric Adenocarcinoma: Glucose vs. Other Energy Sources Substantially Influence Prognosis and Therapy Response” conclude that if diffuse type of gastric adenocarcinoma (dGAC) used glucose as energy source then the prognosis of the patients was very poor while non-glucose sources improved prognosis.
This retrospective analysis demonstrates has limitations:
If they compare low SUV group vs high SUV group - the patients in the groups should be matched and equalized. In the low SUV group are much more patients included with an early stage of gastric cancer. Of course, this has also a major impact on survival. We have no information about molecular and immunological aspects of the gastric cancer (HER2neu, MSS, etc). Additional we need to know, how many patients had pre- and postoperative chemotherapy and what kind of chemotherapy and the number of cycles. We also have to know about the postoperative histopathology (TNM and response shown in downsizing and downstaging). All these information have also an impact on prognosis in gastric cancer patients. If all the information is present - a uni- and multivariate analysis will have a major improvement of the manuscript.
In conclusion, this paper is recommended for the publication in Cancers , but with major concerns
Reviewer 3 Report
Regarding you study, the results of which you present in the current manuscript, I have the following comments:
- I miss a statement that the study had been approved by the competent ethics committee. This is particularly important because patients were exposed to additional radiation during PET without any clear clinical decision based on the exam. Was there a formal study protocol and were patients prospectively included into the study providing informed consent? Or was PET done completely at random without any clear diagnostic and therapeutic or scientific purpose, which I would find ethically questionable. Unfortunately, the methods section provides no information on this.
- One of the major limitations of this study is that FDG-PET was apparently done only in selected patients. Please provide the overall number of patients with gastric cancer, diffuse gatric cancer, and signet ring cell carcinoma treated during the study period and make a comparison regarding important demographic and tumor-specifid parameters. How were patients selected to undergo PET?
- The stratification of tumor site in Siewert Type 3 and stomach seems a bit too unspecific. Could you provide a further sub-differentiation?
- Could you provide some details on the neoadjuvant treatment the patients received. How many underwent chemoradiotherapy and chemotherapy, respectively? What were the chemotherapy schemes and radiation doses applied?
- I couldn't see figures 3-6 in the provided manuscript, so I was unable to assess them.
- Please explain the rationale for using an SUV of 3.5 as a threshold. Are there any data on which this can be based?
- Do you have any information on Her2neu-status of the tumors?
- In the discussion, you state that "The incidence of poorly differentiated tumors and dGACs has been rising[4]. It is interesting that dGAC incidence has been rapidly rising in Blacks for reason that are entirely unclear[4]". You need to point out that this refers to the US, at least if you cite reference [4].
- The language in the second paragraph of the discussion sounds rather unscientific (e.g. "glucose as fuel") and should be improved.
- In the discussion, you state that your study, in conjunction with the study by Chon et al., shows that "[..] the aggressive nature of glucose consuming dGAC remains unaltered whether they have extensive preoperative therapy (our population) or they undergo surgery first[22]. This study is by no means supported by your results, which don't provide any information on the utility of preoperative chemotherapy compared to upfront surgery for dGAC. It should be omitted.
- I miss a circumscribed conclusion drawn from your results. Do your findings have any clinical implication? Should I treat patients with a high SUV differently from those with a low one? You write that "All dGACs appear to have inherent resistance." Resistance to what? And how can such resistance be based on the results of a cross-sectional assessment of FDG avidity in PET in a population of patients who all received preoperative therapy?
Reviewer 4 Report
The authors present a retrospective study comparing diffuse gastric adenocarcinomas based on SUV to pre-therapeutic PET scan. The results show that patients with an SUV less than 3.5 have a better prognosis than those with an SUV greater than 3.5.
Major comment
The detail of patient "race" does not appear to be essential to the understanding of the results or the external validity of the study.
The authors should explain in more detail why they chose an SUV cut-off of 3.5 .
It would be interesting to detail the outcome of the surgical treatment.
Round 2
Reviewer 2 Report
Manuscript improves only slightly, but the information and content is not very new. Therefore I cannot recommend this manuscript to be published in Cancers
Author Response
Manuscript improves only slightly, but the information and content is not very new. Therefore I cannot recommend this manuscript to be published in Cancers
Answer: We added all new tables and the study flowchart to the manuscript and we worked on the discussion part as well.
Reviewer 3 Report
Unfortunately, you have failed to provide sufficient answers to the issues I raised in my first review. Some points have been addressed in the response to reviewers, but not in the manuscript file. I strongly suggest the study flowchart and the different new tables into the manuscript. Referring to your study dataset is not sufficient. Moreover, the objections I had regarding the conclusions you derive from the results, were not answered with scientific arguments. In my opinion, you continue to draw conclusions which are not based on your data.
Author Response
Unfortunately, you have failed to provide sufficient answers to the issues I raised in my first review. Some points have been addressed in the response to reviewers, but not in the manuscript file. I strongly suggest the study flowchart and the different new tables into the manuscript. Referring to your study dataset is not sufficient. Moreover, the objections I had regarding the conclusions you derive from the results, were not answered with scientific arguments. In my opinion, you continue to draw conclusions which are not based on your data.
Answer: We included all new tables and the study flowchart in the revised version of the manuscript. We acknowledge that our findings are by no means conclusive and require validation as our entire cohort is small. Our point is that we should now focus on the molecular biology of these tumors to discover unique targets to improve outcomes because the current approaches are not working.